# Facile transformation of imine covalent organic frameworks into ultrastable crystalline porous aromatic frameworks

Xinle Li[1], Changlin Zhang [1], Songliang Cai[2], Xiaohe Lei[1,3], Virginia Altoe[1], Fang Hong[4], Jeffrey J. Urban[1], Jim Ciston[1], Emory M. Chan[1] & Yi Liu [1]

The growing interest in two-dimensional imine-based covalent organic frameworks (COFs) is inspired by their crystalline porous structures and the potential for extensive π-electron delocalization. The intrinsic reversibility and strong polarization of imine linkages, however, leads to insufficient chemical stability and optoelectronic properties. Developing COFs with improved robustness and π-delocalization is highly desirable but remains an unsettled challenge. Here we report a facile strategy that transforms imine-linked COFs into ultrastable porous aromatic frameworks by kinetically fixing the reversible imine linkage via an aza-Diels-Alder cycloaddition reaction. The as-formed, quinoline-linked COFs not only retain crystallinity and porosity, but also display dramatically enhanced chemical stability over their imine-based COF precursors, rendering them among the most robust COFs up-to-date that can withstand strong acidic, basic and redox environment. Owing to the chemical diversity of the cycloaddition reaction and structural tunability of COFs, the pores of COFs can be readily engineered to realize pre-designed surface functionality.

[1] The Molecular Foundry, Lawrence Berkeley National Laboratory, Berkeley, CA 94720, USA. [2] School of Chemistry and Environment, South China Normal University, 510006 Guangzhou, China. [3] Department of Chemistry, Zhejiang University, 310027 Hangzhou, China. [4] The Advanced Light Source, Lawrence Berkeley National Laboratory, Berkeley, CA 94720, USA. Correspondence and requests for materials should be addressed to Y.L. (email: yliu@lbl.gov)

Covalent organic frameworks (COFs) are porous, crystalline networks constructed by linking molecular organic building units with covalent bonds[1]. Owing to their high surface area, chemical diversity, and tunable functionality, COFs have been adopted as an auspicious platform for a plethora of applications pertaining to gas adsorption[2], separation[3], chemical sensing[4], catalysis[5], optoelectronics[6], and energy storage[7]. In order to establish a covalently linked periodic framework in COFs, current methods rely on the reversible covalent bond formation to connect multivalent monomers through thermodynamic equilibria[8,9]. This inherent reversibility of the linkages within COFs, however, severely limit fundamental materials properties such as environmental stability towards solvents and chemicals, thus constraining their practical applications. For example, COFs with boroxine linkages are susceptible to water or protic solvents[10–12]. COFs based on a more robust linkage, such as the C=N imine bond, represent the prevalent class of COFs that show improved hydrothermal stability[13]. The chemical stability of most imine COFs is however still far from satisfactory since they undergo hydrolysis under strongly acidic conditions or exchange with amines due to the reversible nature of imine[14]. As such, a facile methodology that enables the fabrication of chemically robust COFs is much desired[15–19].

One specific perspective that motivates the field of COFs is the potential of constructing a periodic conjugated 2D framework with covalently linked aromatic ring systems, from which exceptional electronic and magnetic properties are predicted[20]. The imine linkage has been deemed as a better choice than other dynamic bonds as it serves as an $sp^2$-hybridized unit readily inserted into a π-extended framework to open up conjugation pathways. The resulting framework, however, is inferior in facilitating π electron delocalization between the linked units on account of the strongly polarized nature of the C=N bonds[21]. Despite a few examples of crystalline π-extended 2D polymer layers obtained from the surface- or interface-assisted synthesis[22,23], and a most recent $sp^2$ carbon-conjugated COF[24], a general strategy to reinforce strong covalent bonding and extended π-electron delocalization in the same framework remains largely unexplored.

Herein, we explore a strategy that can transform the dynamic imine linkages in COFs into more robust and conductive bonds while preserving the topology, crystallinity, and porosity of COFs. Starting from an imine COF can bypass the crystallization problem encountered in irreversible crosslinking, while the linkage transformation can significantly boost the chemical stability and electron delocalization in the resultant COFs. A number of COFs have been modified postsynthetically[25–28] to give functionally diverse and topologically identical porous materials. The modification reaction happens almost exclusively on the linkers that bear chemically addressable pendant groups, with only one exception in which the reaction targets the imine linkages to transform them into more stable amide linkages[29]. Herein, we report a more versatile approach to kinetically fix the reversible linkage in imine COFs to afford crystalline and chemically robust porous 2D aromatic framework with extended π-electron delocalization (Fig. 1a). Thanks to the efficient Povarov (aza-Diels-Alder, aza-DA) reaction[30] between aryl imines and arylalkynes, the imine linkages in 2D COFs can be converted to yield the corresponding quinoline-linked COFs (denoted as MF-1a-e). Owing to the structural tunability of starting imine-linked COFs and substrate diversity of the aza-DA reaction, a large variety of functional moieties can be introduced to selectively alter the pore surface and wettability of the resultant COFs.

## Results and discussion

### Synthesis of the MFs via aza-DA reaction. The viability of the Povarov reaction was first demonstrated on a prototype imine-

linked COF, a highly crystalline and porous COF denoted as COF-1[31]. COF-1 was readily obtained following a reported protocol and was subjected to the reaction with phenylacetylene (**1a**) at 110 °C in the presence of BF₃•Et₂O (1.5 equiv. per imine functionality) and chloranil in toluene for 72 h. The solid separated from the reaction mixture was washed with excess anhydrous THF and saturated NaHCO₃, and then dried under vacuum to afford MF-1a as dark yellow solid.

**Characterizations of the MFs**. The successful addition of phenylacetylene onto COF-1 was verified by several analytical methods. Comparison of the Fourier-transform infrared (FT-IR) spectra of MF-1a and COF-1 (Fig. 2a) revealed a new cluster of peaks at ~1600 cm⁻¹ that correspond to the stretch of aromatic quinoline core[30] and blue shift of the C–C=N-C stretch (from 1202 to 1222 cm⁻¹) after the reaction. As a control, the 1D imine-polymer (Supplementary Figure 1) was subjected to Povarov reaction to afford modified quinoline-linked polymer, which revealed a similar cluster of peaks at ~1600 cm⁻¹ arising from the aromatic quinoline core and blue shift of the C-C=N-C stretch (from 1207 to 1214 cm⁻¹). Moreover, the C-C=N-C stretching is close to the model compound, diphenyl quinoline (1238 cm⁻¹). Another control experiment run under the identical reaction conditions except without phenylacetylene **1a** confirmed that the Lewis acid catalyst (BF₃•Et₂O) and chloranil alone induced no chemical modification of COF-1, as revealed by the lack of evident changes in the FT-IR spectra of COF-1 (Supplementary Figure 2). Given that the quinoline vibration at ~1622 cm⁻¹ in the spectrum of MF-1a overlaps with C=N imine stretch of COF-1, **1a** was replaced with a *para*-ester substituted phenylacetylene derivative (**1b**) to afford MF-1b bearing a characteristic chemical probe. FT-IR spectra of MF-1b revealed a peak at ~1725 cm⁻¹, arising from the carbonyl moiety (Fig. 2a), which is also different from that in the spectra of free **1b** (~1702 cm⁻¹) (Supplementary Figure 3). Solid-state ¹³C cross polarization magic angle spinning (CP-MAS) NMR spectroscopy revealed an apparent up-field shift of the aromatic and -OMe resonances in the spectrum of MF-1a compared to that of COF-1. Such up-field shift is again observed in the spectra of MF-1b, together with two well-resolved but characteristic peaks at 165 and 50 ppm that could be assigned to the methyl ester groups (Fig. 2b). These spectroscopic changes corroborate well with successful conversion of the imine linkages of the COF framework following the Povarov cycloaddition.

The powder X-ray diffraction (PXRD) pattern of COF-1 exhibited six prominent diffraction peaks, with the most intensive one at 2.73° and the five other peaks at 4.79°, 5.54°, 7.35°, 9.65°, and 25.2°, assigned to the (100), (110), (200), (210), (220), and (001) facets, respectively (Fig. 2c, black curve). After modification, the PXRD pattern of MF-1a exhibited four prominent diffraction peaks, with the most intensive one at 2.75° and the three other peaks at 4.82°, 5.58°, and 25.1°, corresponding to the (100), (110), (200), and (001) facets, respectively (Fig. 2c, red curve). The (100) peaks of MF-1a and MF-1b have a small full-width at half-maximum (FWHM) values of 0.25° and 0.29°, respectively, suggesting their high crystallinity. Such peaks show insignificant shift compared to that of COF-1 (FWHM = 0.24°) (Fig. 2c), indicating preservation of the crystalline framework during the linkage transformation. Using an optimized monolayer structure, AA and staggered AB stacking modes were generated and optimized. The simulated PXRD pattern of the modeled 2D frameworks in AA stacking mode is in good agreement with the experimental peak positions. The PXRD pattern of MF-1a and MF-1b demonstrated similar high crystallinity and insignificant shift of (100) and (001) peaks as compared with that of COF-1 (Fig. 2c), indicating the preservation of the crystalline framework

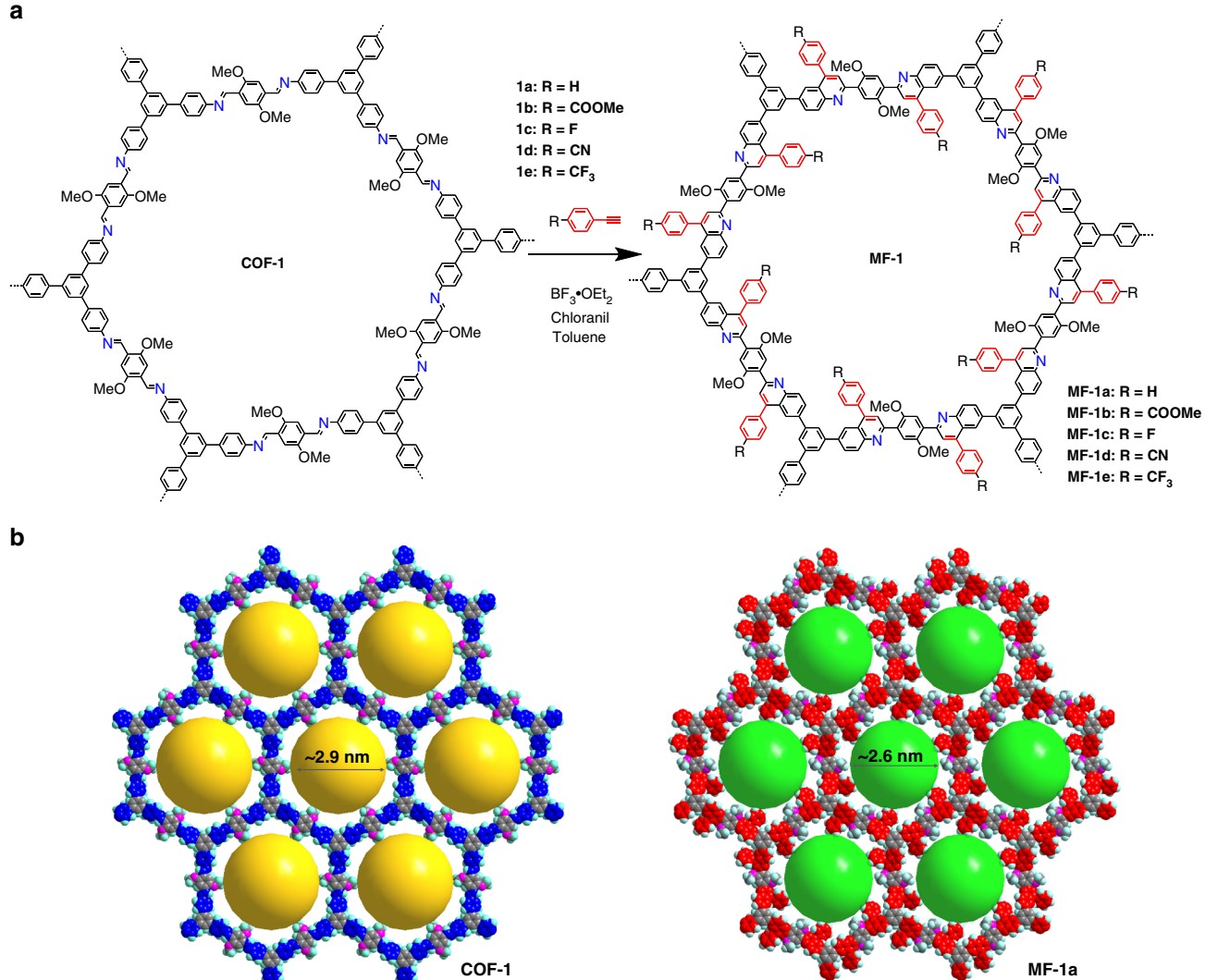

**Fig. 1** Post-synthetic modification of COFs via aza-DA reaction. **a** The reaction scheme showing the transformation of one lattice unit of COF-1 into MF-1a-e. **b** The simulated modeled structure showing the extended frameworks of COF-1 and MF-1a. The yellow and cyan sphere indicate pore diameter of ~2.9 and ~2.6 nm. Note that the illustrated structure of MF-1 contains fully converted quinolines and does not represent the actual degree of transformation

during the linkage transformation. Such patterns also matched with the simulated PXRD pattern of the modeled 2D frameworks in AA stacking mode (Fig. 1b and Supplementary Figure 4c). Pawley refinement showed the negligible difference between the simulated XRD and experimental patterns (Supplementary Figure 4b), while the staggered AB stacking mode (Supplementary Figure 4d) did not reproduce the experimental profile (Supplementary Tables 1–5). UV-Vis diffuse reflectance spectra of MF-1a (Supplementary Figure 5a, red curve) showed a red shift as compared with COF-1 (Supplementary Figure 5a, black curve), which is indicative of enhanced π delocalization. Upon modification, MF-1a showed a reduced optical bandgap of 2.30 eV compared to that of COF-1 (2.52 eV). As shown by thermogravimetric analysis (Supplementary Figure 6), MF-1a has a slightly higher decomposition onset temperature compared to COF-1. In addition, it displays less weight loss than COF-1 after incubation at 400 °C, indicating better thermal stability after the chemical transformation.

$N_2$ sorption analyses performed at 77 K revealed a Brunauer−Emmett−Teller (BET) surface area of 1760, 955, and 590 $m^2\,g^{-1}$ for COF-1, MF-1a, and MF-1b, respectively (Fig. 2d). The decrease of surface area corroborates with the increased framework mass and reduction of the pore volume. The type IV

isotherms of MF-1a, MF-1b, and COF-1 are not only indicative of the mesoporous characters, but also suggest that the chemical modification does not lead to significant changes in the framework structure. Pore size distribution analysis indicated a pore size reduction from 3.0 to 2.1 nm upon chemical transformation (Supplementary Figure 7). Additional SEM studies showed that the morphology of MF-1a is similar to that of the pristine COF-1 (Supplementary Figure 8). X-ray photoelectron spectroscopy (XPS) analyses were performed to provide more insight into the conversion from imine to quinoline. As shown in Supplementary Figure 9, the N1s peak at ~398.0 eV in COF-1 can be attributed to the imine N atoms[32]. Upon modification, a fraction of the N1s core level is shifted to higher binding energies (~399.5 eV), corresponding to the C=N in quinoline moieties[33]. Based on the integration of the peak areas, the degrees of functionalization in MF-1a and MF-1b are assessed to be 27% and 25%, respectively. Further XPS depth profiling analysis indicates that the quinoline/imine nitrogen ratio remains constant after repeated exposure to high energy argon ion beam irradiations, confirming a uniform material composition from the surface to the buried body of the bulk (Supplementary Figure 10). In the case of MF-1a, increasing the reaction time to 1 week resulted in a slight increase of the conversion to 29%, while increasing the concentration of all the

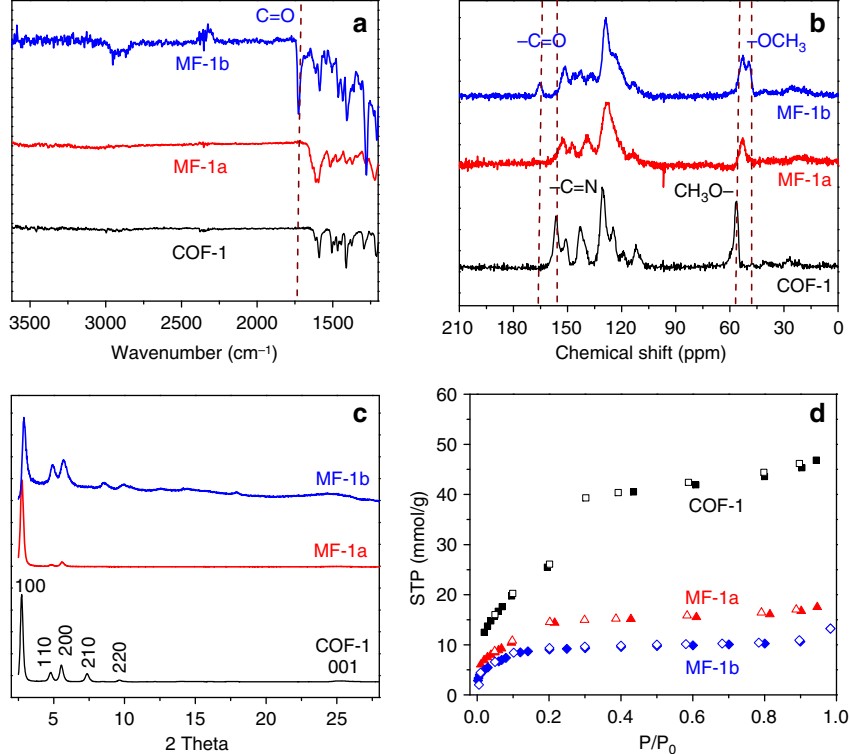

**Fig. 2** Characterization of original COF-1 (black) and post-synthetically modified MF-1a (red) and MF-1b(blue). **a** FT-IR spectra. **b** Solid-state $^{13}$C CP-MAS NMR spectra. **c** PXRD patterns showing retention of crystallinity after the Povarov reaction. **d** N$_2$ sorption isotherm curves

reactants except COF-1 by 3-fold augmented the conversion to 35% (Supplementary Figure 11), suggesting that higher conversion may be achieved by further optimization of the reaction conditions.

The periodic framework structural features of COF-1 and MF-1a were visualized by high-resolution transmission electron microscopy (HRTEM), using the low-dose TEM technique equipped with Gatan K2 Summit direct-detection electron-counting camera[34] and geometric/chromatic image aberration correction. The total electron dose was 2.1 e Å$^{-2}$ and the pixel size 0.72 Å. As shown in Fig. 3a, the honeycomb-like porous structure of COF-1 was clearly observed along the [111] direction, with the pore opening determined as 3.0 ± 0.2 nm. Based on the Fourier-filtered image of a selected area (red square in Fig. 3a and Supplementary Figure 12), we can clearly observe a hexagonal projected symmetry constructed by six white diffraction spots. Upon modification, MF-1a shows a decreased pore opening of 2.3 ± 0.2 nm (Fig. 3c) with the preservation of similar honeycomb-like porous structure (Fig. 3b, d). The direct observation of the pore opening decrease in the framework structures before and after modification, which is consistent with the modeled structures as shown in Fig. 1b and pore size distributions (Supplementary Figure 7), confirmed the effectiveness of the framework-to-framework transformation. Note that COF samples are highly electron-beam sensitive, making the acquisition of high-resolution extremely challenging[35–37], this work represents a rare example in which the individual building units of COFs are directly and clearly observed by HRTEM.

**Chemical stability**. To assess the chemical stability of the modified framework structure, MF-1a was subjected to a variety of extremely harsh chemical conditions. The post-synthetically

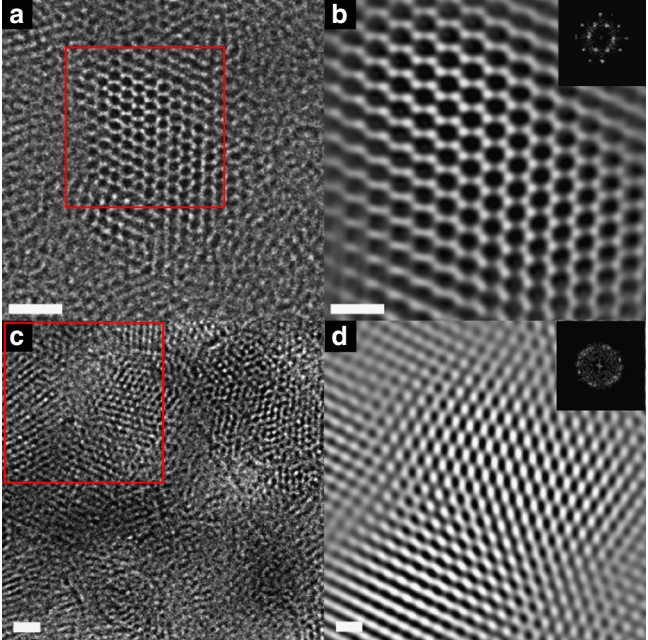

**Fig. 3** HRTEM characterization of COF-1 and MF-1a. **a** Low-dose, high-resolution TEM image of COF-1 (scale bar, 10 nm). **b** The Fourier-filtered image of selected red square areas (scale bar, 5 nm), Inset: Fast Fourier Transform (FFT) from the red square on the COF-1. **c** Low-dose, high-resolution TEM image of MF-1a (scale bar, 10 nm). **d** The Fourier-filtered image of the selected red square area (scale bar, 5 nm), Inset: FFT from the red square on the MF-1a

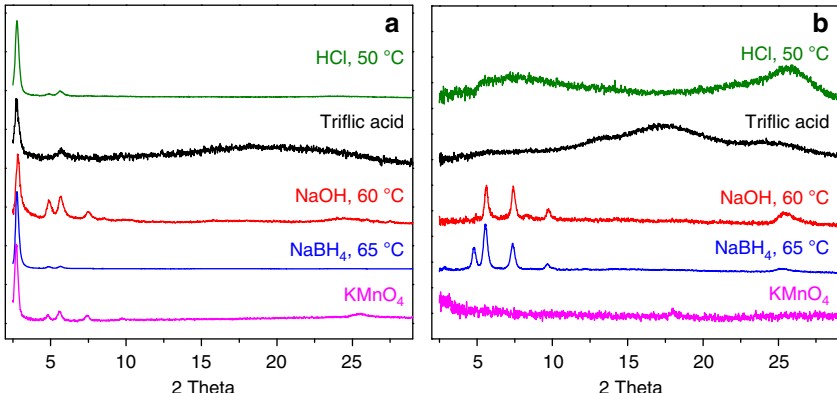

**Fig. 4** Chemical stability of MF-1a and COF-1. PXRD patterns of MF-1a (**a**) and COF-1 (**b**) after treatment with 12 M HCl at 50 °C for 8 h (green), 98% triflic acid at ambient temperature for 3 days (black), 14 M NaOH in H$_2$O/MeOH solution at 60 °C for 1 day (red), 5 equiv. of NaBH$_4$ in MeOH at 65 °C for 1 day (blue), and 5 equiv. of KMnO$_4$ in H$_2$O/CH$_3$CN solution at ambient temperature for 1 day (purple)

modified COF exhibited strikingly high chemical stability in strong mineral acid (12 M HCl at 50 °C for 8 h), superacid (98% TfOH, 3 days), strong base (14 M NaOH in H$_2$O/MeOH at 60 °C, 1 day), strong oxidant (KMnO$_4$ in H$_2$O/CH$_3$CN solution, 1 day) and reducing agents (NaBH$_4$ in MeOH at 65 °C, 1 day), as revealed by the retention of major diffraction peaks in the PXRD patterns of the treated samples (Fig. 4a). Remarkably, MF-1a retained its crystallinity after being exposed to 12 M HCl at room temperature for 2 months (Supplementary Figure 13), or to boiling acids (1 M and 12 M HCl) and bases (1 M and 14 M NaOH) for 1 day (Supplementary Figure 14). IR spectra of the acid-treated MF-1a indicated the appearance of aldehyde vibration band at ~1670 cm$^{-1}$, suggesting partial hydrolysis of the remaining imine bonds under such forcing conditions (Supplementary Figure 15). As the porous framework is well preserved due to the introduced quinoline linkage, the hydrolysis conceivably gives a patchy framework that is decorated with dangling aldehyde and amine functional groups, which present further opportunities for introducing extra functionality. Such a high framework stability compares favorably to other known stable framework materials (Supplementary Table 6). In sharp contrast, the corresponding COF-1, one of the most chemically robust imine COFs reported up-to-date, is rendered amorphous or loses framework periodicity (Fig. 4b and Supplementary Figures 13, 14). The residual weight percentages of MF-1a under the conditions of strong acid (12 M HCl at 50 °C) and strong base (14 M NaOH at 60 °C) are 88 and 83 wt% while the pristine COF-1 exhibited more significant weight loss after acid/base treatment (Supplementary Figure 16). Moreover, MF-1a still remained porous with only 5–25% decrease in surface area. whereas the pristine COF-1 displayed much more significant loss (80–100%) in surface area (Supplementary Figure 17). The vast differential in the stability of the MF-1 library relative to COF-1 clearly underscores the significance of transforming dynamic imines to much more robust quinoline units. The greatly enhanced stability towards chemical oxidation and reduction is particularly remarkable and is relevant for practical applications involving redox processes.

**Generality of the Povarov addition approach**. The cycloaddition-based approach allows the integration of various functionalities onto the pores of COFs from substituted arylalkynes, thus rendering it much more versatile than the oxidation-based post-synthetic modification of imine-COFs[29], and click reaction-based post-synthetic modification of COFs, which requires specifically designed azide/ethynyl-appended building blocks and metal catalyst (e.g., CuI)[25,26]. The generality was demonstrated by reacting COF-1 with *para*-substituted arylalkynes **1c–e** under similar reaction conditions to give MF-1c-e bearing fluorine, nitrile and trifluoromethyl groups on the pore surfaces. The PXRD analysis confirmed the retention of crystallinity after the Povarov reaction (Fig. 5a) while solid-state CP-MAS $^{13}$C NMR (Supplementary Figure 18) and FT-IR spectroscopy revealed changes that correspond to the conversion of imine bonds to quinoline C=N bonds (Fig. 5b). Notably, the characteristic -CN vibrational peak (~2231 cm$^{-1}$) and -CF$_3$ stretch (~1325 cm$^{-1}$) in the FT-IR spectra of MF-1b and MF-1c, which are different from these in the spectra of free **1d** (2226 cm$^{-1}$) and **1e** (1320 cm$^{-1}$) (Supplementary Figure 19), together with the absence of alkyne peaks (~3200–3300 cm$^{-1}$ for -C≡C-H stretch and ~2100 cm$^{-1}$ for -C≡C- stretch), unambiguously confirmed the chemical attachment of -PhCN and –PhCF$_3$ groups to the framework (highlighted in cyan in Fig. 5b) that was concomitant with successful Povarov reaction. XPS analysis shows the degree of functionalization for MF-1 c-e is 27%, 29%, and 26%, respectively (Supplementary Figure 9), very similar to that of MF-1 a-b.

This post-synthetic modification protocol can be applied to other imine-COFs as well, as demonstrated in the reaction with COF-2 (Supplementary Figure 20), which was readily synthesized following a routinesolvothermal procedure[38]. The disappearance of the characteristic imine stretch at 1625 cm$^{-1}$ is clearly discernible in the FT-IR spectra of MF-2a (Supplementary Figure 21). The retention of morphology, crystallinity, and porosity in MF-2a was confirmed by SEM images (Supplementary Figure 22), PXRD analysis (Supplementary Figure 23) and N$_2$ sorption analysis (Supplementary Figure 24), respectively. The enhanced chemical stability was demonstrated by treating MF-2a with a strong acid (12 M HCl, 1 day), the PXRD of which remained unchanged while COF-2 decomposed completely under the same conditions (Supplementary Figure 25). Finally, a variety of organic functionalities such as -F, -CN, and -CF$_3$ groups have been successfully incorporated into the porous crystalline framework via the Povarov reaction of COF-2, as verified by PXRD and FT-IR analyses (Supplementary Figure 26). XPS analysis indicates the degree of functionalization in MF-2 a-d is 21%, 29%, 24%, and 29%, respectively (Supplementary Figure 27), which is comparable to the MF-1 series. All these results correlate well with a successful chemical transformation of imine-COFs into more stable crystalline porous frameworks.

**Customizable surface wettability**. The customizable surface functionality of stable COFs provides a convenient way to

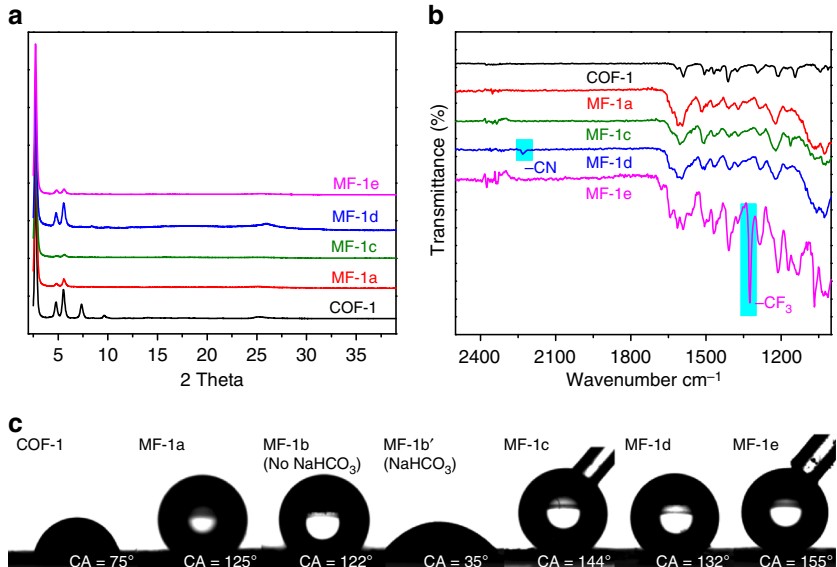

**Fig. 5** Structural characterization and surface properties of MF-1 series. **a** Powder XRD patterns of COF-1 (black), MF-1a (red), MF-1c (green), MF-1d (blue), and MF-1-e (purple). **b** FT-IR spectra of COF-1 and MF-1a, c-e. The peaks highlighted in cyan are characteristic vibrations from these functional groups. **c** Water contact angles (CA) of water droplet on the pressed pellet of COF-1 and MF-1a-e. MF-1b' is the NaHCO₃-treated MF-1b that undergoes partial ester hydrolysis

fabricate porous materials with controlled wettability, which has garnered continuous attention from both academia and industry[39–41]. The surface wettability of COFs before and after the Povarov reaction differs drastically and shows a strong dependence on both the COF structure and the substituent groups on the phenylacetylene. Figure 5c shows water contact angles on the surface of COF-1 and MF-1a-e. The contact angle on MF-1a is ~125° while COF-1 exhibits a contact angle of only ~75°, indicating that the covalent modification of COFs significantly enhanced the hydrophobicity of the material. This increase in hydrophobicity is presumably due to the introduction of more hydrophobic aromatic rings onto the pore surfaces and decreased polarization after incorporating the imine bonds into the aromatic quinoline ring systems. Notably, incremental changes of contact angles are observed when altering the functional groups from –H (MF-1a, 125°) to –CN (MF-1d, 132°), –F (MF-1c, 144°), and –CF₃ (MF-1e, 155°), highlighting an effective way to systematically fine-tune COF surface wettability into the superhydrophobic region. The methyl ester-bearing MF-1b has a similar contact angle of 122°. Upon treatment with NaHCO₃, the contact angle (MF-1b′) decreases significantly to 35°, indicating the feasibility of imparting hydrophilicity via hydrolysis of ester moieties. Similar fine modulation of hydrophobicity is also observed in the series of MF-2s. MF-2a has a contact angle of 152° which is significantly higher than the 125° contact angle of the pristine COF-2 and the introduction of fluorinated functional groups (–F, and –CF₃) led to gradually increased hydrophobicity, evidenced by the contact angle change from –H (MF-2a, 152°) to –F (MF-2c, 157°) and –CF₃ (MF-2e, 162°) (Supplementary Figure 28).

We have developed a facile approach to deriving crystalline porous aromatic frameworks from readily available imine-COFs by transforming the dynamic imine linkages into more stable quinoline aromatic ring systems via aza-Diels-Alder cycloaddition. The kinetic fixation of the imine linkages, even with only around 20–30% conversion, resulted in dramatically enhanced framework stability towards strong acid, base, and redox reagents. This framework-to-framework transformation offers a simple solution to the intrinsic instability associated with imine-COFs

while retaining the framework's crystallinity and permanent porosity, while simultaneously enabling the tuning of pore surface functionalities and π electron delocalization. This succinct protocol paves the way to the synthesis of crystalline, porous aromatic frameworks that are difficult to obtain de novo, and it will facilitate practical applications of organic framework materials that require enhanced chemical stability, semiconducting properties, and pore surface functionality.

## Methods

**Reagents.** Acetone, acetonitrile, chloroform, dichloromethane, methanol, tetrahydrofuran (THF), toluene, and 1,4-dioxane were purchased from Fisher chemicals. 1-ethynyl-4-fluorobenzene and 1,3,5-tri-(4-aminophenyl)benzene were purchased from TCI. 1-ethynyl-4-(trifluoromethyl)benzene as purchased from 1Click Chemistry Stock Products. Methyl 4-ethynyl benzoate was purchased from AK scientific. 2,5-Dimethoxyterephthalaldehyde were purchased from Carbosynth chemicals. 2,4-diphenyl-quinoline were purchased from EnamineStore. Phenylacetylene, 4-ethynyl benzonitrile, boron trifluoride diethyl etherate, triethylamine were purchased from Alfa Aesar chemicals. Chloranil, triflic acid, 1,3,5-triformylbenzene, benzidine, mesitylene, o-dichlorobenzene, n-butanol were purchased from Sigma Aldrich chemicals.

**Instrumentation and characterization.** Fourier transform infrared (FT-IR) spectra were recorded on a Perkin Elmer Spectrum One FT-IR system. Powder X-ray diffraction (PXRD) patterns were recorded on a Bruker Discovery D8 X-ray diffractometer and Rigaku MiniFLex 6G Benchtop XRD with Cu Kα1 radiation (λ = 1.5406 Å). Nitrogen sorption isotherms were obtained at 77 K with a Micromeritics Instrument Corporation model 3Flex surface characterization analyzer. The Brunauer-Emmett-Teller (BET) method was utilized to calculate the specific surface areas. By using the non-local density functional theory (NLDFT) model, the pore size distribution was derived from the sorption curve. TGA measurements were performed on a TA Instruments Q5000IR TGA under Argon, by heating to 600 °C at a rate of 10 °C min⁻¹. Solid-state ¹³C CP-MAS NMR spectra were recorded on a BrukerAvance500 I (MF-1d sample only) and BrukerAvance500 II. The UV-vis diffuse reflectance measurement was performed on Cary 5000 UV-Vis-NIR spectrometer. X-ray photoelectron spectroscopy (XPS) measurement was carried out on a Thermo Scientific K-Alpha XPS apparatus equipped with a monochromatic Al K(alpha) source and food gun for charge compensation. To remove the acid residues, the COFs samples were treated with triethylamine methanolic solution, followed by washing with copious amount of methanol and degas at 200 °C under dynamic vacuum for 1 day prior to XPS measurement. Low-dose TEM images of COF-1 were acquired on the TEAM I FEI Titan-class microscope at 300 kV, equipped with both geometric aberrations corrected to third order and chromatic aberrations corrected to the first order. Imaging data were collected in the Gatan K2 direct-detection camera operated in electron-counting

mode (camera counting frame rate of 400 fps (frames per second) at $4\,k \times 4\,k$ resolution) with a final image output rate of 40 fps at $4\,k \times 4\,k$ resolution. The HRTEM image of MF-1a was acquired with an FEI TitanX 60 300 microscope at 200 kV. The COF sample was sonicated in toluene with a sonication probe for 15 min and drop-casted onto a copper grid (Lacey C only, 300 mesh Cu). SEM images were obtained with a Zeiss Gemini Ultra-55 Analytical Field Emission Scanning Electron Microscope operated at 15 kV using an in-lens detector. Water contact angle measurements were carried out using a Kruss easy drop optical contact angle meter (Model: FM41) under ambient conditions.

**Crystal structure modeling**. Since the framework structures of MF-1a-e are quite similar, and those of MF-2a-e are close as well, thus only MF-1a and MF-2a are selected as representative examples for structure modeling. The crystal models for MF-1a and MF-2a including the cell parameters and the atomic positions were produced by Materials Studio 5.0 software package1 employing the Crystal Building module. The Pawley PXRD refinement was conducted using the Reflex module in the Materials Studio 5.0, in which a Pseudo-Voigt profile function was employed for whole profile fitting (peak broadening, peak asymmetry, and zero shift error were taken into account). Unit cell and sample parameter were refined simultaneously. The Pawley refinement results including unit cell parameters and final related refinement factors for MF-1a and MF-2a were listed in Supplementary Table 1, while the atomic coordinates of MF-1a and MF-2a crystal structure models are presented in Supplementary Tables 2–5.

**Synthesis of COF-1**. An o-dichlorobenzene/n-butanol (1 mL/1 mL) mixture of 1,3,5-tri-(4-aminophenyl) benzene (56 mg, 0.16 mmol) and 2,5-dimethoxy ter-ephthalaldehyde (46 mg, 0.24 mmol) in the presence of acetic acid (6 M, 0.2 mL) in a Biotage microwave vial (5 mL) was degassed through 3 freeze–pump–thaw cycles. The vial was sealed and heated at 120 °C for 3 days. The precipitate was collected via centrifugation, washed times with anhydrous THF and subjected to Soxhlet extraction using THF as the solvent for 1 day. The powder collected was dried at 120 °C under vacuum overnight to give yellow colored COF-1 in an isolated yield of ~80%.

**Synthesis of COF-2**. Benzidine (64 mg, 0.2 mmol) and 1,3,5-triformylbenzene (32 mg, 0.15 mmol) were dissolved via sonication in a mixture of 1,4-dioxane/mesitylene (1 mL/1 mL) in a 5 mL-Biotage microwave vial. Afterwards, aqueous acetic acid (6 M, 0.2 mL) was added to the mixture. The vial was degassed by three freeze–pump–thaw cycles. Finally, the vial was sealed and heated in an oven at 120 °C for 3 days. The precipitates were isolated by centrifugation, washed with anhydrous THF for three times, and dried at 120 °C under vacuum overnight to give a yellow colored powder in ~90 % yield.

**Synthesis of MFs via Povarov reaction**. COFs (4 mg), phenylacetylene (6 µL, 0.05 mmol), BF₃•OEt₂ (4 µL, 0.03 mmol), chloranil (8 mg, 0.03 mmol), and 2 mL of toluene were added to a 5 mL-Biotage microwave vial. The vial was sealed and heated under $N_2$ at 110 °C in an oil bath. After 1–3 days, the mixture was cooled to room temperature and the precipitate was isolated via centrifugation. The reaction mixture was then washed with THF and quenched with saturated aqueous sodium bicarbonate (2 mL × 3). Subsequently, the solids were washed with THF using a Soxhlet extractor for 12 h and dried under vacuum.

**Chemical stability test of MF-1a**. The MF-1a sample (~2 mg) was kept for pre-designed time under static condition in 0.5 mL of 98% triflic acid (2% catalytic $H_2O$) at ambient temperature, 0.5 mL HCl (1 and 12 M) at ambient temperature, 50 °C or 100 °C, NaOH (1 and 14 M) in $H_2O$/methanol solution at 60 °C or 100 °C, NaBH₄ (5 equiv. per imine functionality in MF-1a) in methanol at 60 °C and KMnO₄ (5 equiv. per imine functionality in MF-1a) in $H_2O$/CH₃CN solution. The samples were washed with THF for three times, dried under vacuum at ambient temperature and subjected to PXRD measurements.

**Data availability**. All relevant data supporting the findings of this study are available from the corresponding author on request.

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

## Acknowledgements

Work at the Molecular Foundry was supported by the Office of Science, Office of Basic Energy Sciences, of the U.S. Department of Energy under Contract No. DE-AC02-05CH11231. S.C. is grateful to the support from the National Nature Science Foundation of China (Grant No. 21603076). We thank Liana M. Klivansky and Haoran Yang for their help in structural characterizations of our samples.

## Author contributions

Y.L., E.C., J.C., and J.U. conceived the project and provided funding. Y.L. designed the experiments, X.Li and X.Lei conducted the experiments, C.Z. performed TEM measurement and analysis, S.C. performed PXRD simulations and analysis, V.A. assisted with XPS measurements, F.H. assisted with PXRD measurements and Y.L., E.C., J.C., J.U., and X.Li wrote the manuscript.

## Additional information

**Competing interests:** A patent application has been filed before the submission of this manuscript. Apart from that, the authors declare no competing interests.

