## [Peer Review File · Nature Communications]

Reviewers' comments:

Reviewer #1 (Remarks to the Author):

For the response of comment 8, reviewer 1:

XPS is a surface-sensitive quantitative spectroscopic technique which does not give the quantitative elemental composition of the bulk materials. Hence the assessment of the degree of functionalization in MF-1a and MF-1b based on the integration of the peak areas in XPS spectra, is not appropriate at all. For calculating the degree of functionalization in MF-1a and MF-1b, quantitative ¹³C solid-state NMR could be a better option.

XPS analysis shows the degree of functionalization for MF-1 and MF-2 series are 20-30% only. With the increase, the reactants concentration by 3 fold the yield has been increased to 35% only (increment of 5% only). That indicating only 35% of the imine bonds present in the crystalline framework have been converted into quinoline moiety and the rest 65% imine bonds are remaining intact. So how the stability of the MF-1 and MF-2 series has been increased to this extent after treatment of 12 M HCl? Also in order to increase the conversion rate, instead of increasing the reactants concentration the author could increase the no of reaction cycle performed on pristine COF-1. After completing the 1st cycle of Povarov reaction, the same reaction could be performed for quite a few cycles with the product of the first cycle (after separating from the reaction mixture) in order to increase the conversion rate into a greater extent.

For the response to comment 2, reviewer 1:

If the author could able to increase the conversion rate of the Povarov reaction, better quality ¹³C NMR spectra could be achieved for the post-synthetically modified COFs. It will be better to have a ¹³C solid state CP-MAS NMR spectra for MF-1d also for the confirmation of the presence of -CN in MF-1d.

For the response to comment 6, reviewer 2:

According to the author, the MF-1 is stable in 12 M HCl at 50 oC and in 14 M NaOH at 60 oC (Table S6). What will happen if the temperature is increased to 100 oC? In case of CAF-1 and CAF-2, they are stable up to 100 oC in HCl and as well as NaOH solution.

Rest of the others comments have been addressed by the author suitably.

In addition, one more point should be addressed by the author.

1. For MF-1b the contact angle is 35o (indicating hydrophilic in nature) and for MF-1d the contact angle is 132o (indicating hydrophobic in nature). According to the literature report (J. Phys. Chem. B 2013, 117, 7718–7723 & Phys. Chem. Chem. Phys., 2014, 16, 13262-13270) through the dipole moment of -CN is higher than -(C=O)O- group but their affinity towards water as well as solubility in water are almost same and they are both hydroneutral in nature. But the contact angle measurement of the MF-1b and MF-1d contradicts this

Reviewer #2 (Remarks to the Author):

I thank the authors for the exemplary revision and additional effort. I recommend this draft for speedy publishing in Nat. Commun.

Reviewer #1 (Remarks to the Author):

1. For the response of comment 8, Reviewer 1:

XPS is a surface-sensitive quantitative spectroscopic technique which does not give the quantitative elemental composition of the bulk materials. Hence the assessment of the degree of functionalization in MF-1a and MF-1b based on the integration of the peak areas in XPS spectra, is not appropriate at all. For calculating the degree of functionalization in MF-1a and MF-1b, quantitative ^{13}C solid-state NMR could be a better option. XPS analysis shows the degree of functionalization for MF-1 and MF-2 series are 20-30% only. With the increase, the reactants concentration by 3 fold the yield has been increased to 35% only (increment of 5% only). That indicating only 35% of the imine bonds present in the crystalline framework have been converted into quinoline moiety and the rest 65% imine bonds are remaining intact. So how the stability of the MF-1 and MF-2 series has been increased to this extent after treatment of 12 M HCl? Also in order to increase the conversion rate, instead of increasing the reactants concentration the author could increase the no of reaction cycle performed on pristine COF-1. After completing the 1st cycle of Povarov reaction, the same reaction could be performed for quite a few cycles with the product of the first cycle (after separating from the reaction mixture) in order to increase the conversion rate into a greater extent.

Response: Regarding XPS and quantitative analysis:

XPS has been widely adopted as a quantitative model for composition analysis, including nitrogen-doped carbon materials and COF materials, both of which are relevant to the current studies. Listed here are two most recent examples from Donglin Jiang, the leading COF researcher who has used XPS for COF composition analysis:

“Ion Conduction in Polyelectrolyte Covalent Organic Frameworks”, *J. Am. Chem. Soc.* **2018**, ASAP, DOI: 10.1021/jacs.8b03814. for Li composition analysis (Figure S3).

“Template Conversion of Covalent Organic Frameworks into 2D Conducting Nanocarbons for Catalyzing Oxygen Reduction Reaction”, *Adv. Mater.* **2018**, *30*, 1706330, for N and P composition analysis (Figure 5).

Two examples of using XPS for nitrogen-doped carbons are also listed:

“Active Sites of Nitrogen-doped Carbon Materials for Oxygen Reduction Reaction Clarified using Model Catalysts”, *Science*, **2016**, *351*, 361-365, for N composition analysis (Figure 3A).

“Mesoporous Nitrogen-rich Carbons Derived from Protein for Ultra-high Capacity Battery Anodes and Supercapacitors”, *Energy Environ. Sci.*, **2013**, *6*, 871–878, for N composition analysis (Table 1).

That being said, we agree that XPS is a surface-sensitive technique. To probe the composition at the buried body of the bulk materials, we have further conducted XPS depth profiling analysis of MF-1a by sputtering the sample surface with high energy argon ion clusters. As shown in XPS depth profile in Figure R1, the composition remains constant after four 60-second exposures at each of two separate ion beam energies (Ar_{1000}^+ at 6 keV and Ar_{75}^+ at 6 keV),

indicating a uniform material below the top surface. Furthermore, the sample uniformity is consistent with the comparative IR measurements using both a transmission and a surface-sensitive ATR configuration, which revealed similar structural features from both methods (please refer to our response to Reviewer 2 in the previous response letter).

Figure R1. (a) N1s XPS spectra of MF-1a before sputtering the sample surface, after sputtering the sample surface with argon ion cluster (Ar_{1000}^+ at 6 keV), and (Ar_{75}^+ at 6 keV) for 4 cycles of 60 s each; (b) Plot of the degree of functionalization against the sputtering cycles.

We thus believe that XPS has provided a reliable approximation of the degree of functionalization. Regarding the suggested quantitative solid-state ^{13}C -NMR measurement, we are not confident that solid-state NMR has enough resolution to differentiate the characteristic chemical shifts between MF-1 and COF-1, as further evidenced by the newly acquired ^{13}C CP-MAS NMR of MF-1d (see below response to Comment 2). We would also like to point out that quantitative solid-state ^{13}C -NMR has yet to be demonstrated practical for COF materials, as there is no prior example of its use for COFs, and is unlikely to yield meaningful outcome for the current work. Corresponding changes are made on page 8:

“Further XPS depth profiling analysis indicates that the quinoline/imine nitrogen ratio remains constant after repeated exposure to high energy argon ion beam irradiations, confirming a uniform material composition from the surface to the buried body of the bulk (Figure S10).”

Regarding the stability of residue imine bond:

The retention of crystallinity after strong acid treatment suggests that despite a partial conversion of imine bonds, the framework is effectively stitched together by the formation of quinoline linkages. We reason that the linkage modification (imine to quinoline) and the boosted hydrophobicity (evidenced by contact angle measurement) contributed to the increased hydrolytic stability in 12M HCl. It is widely reported that the increased hydrophobicity could significantly enhance the stability of porous materials in aqueous media (e.g., *J. Am. Chem. Soc.*, **2012**, *134*, 14338–14340, *J. Am. Chem. Soc.*, **2014**, *136*, 16978–16981). On the other hand, the reappearance of aldehyde signals in IR spectra was observed after treating MF-1a in 1M HCl at 100 °C for 24 hours (Figure R2), suggesting that hydrolysis of the imine bonds does occur under forcing acidic conditions. Nevertheless, the framework is still maintained due to the introduced quinoline linkage. The hydrolysis conceivably gives a “patchy” framework decorated with

dangling aldehyde and amine functional groups, which present further opportunities for introducing extra functionality. More effort in exploring this novel concept is currently ongoing. Corresponding changes are made on page 10:

“IR spectra of the acid-treated MF-1a indicated the appearance of aldehyde vibration band at $\sim 1670\text{ cm}^{-1}$, suggesting partial hydrolysis of the remaining imine bonds under such forcing conditions (Figure S15). As the porous framework is well preserved due to the introduced quinoline linkage, the hydrolysis conceivably gives a “patchy” framework that is decorated with dangling aldehyde and amine functional groups, which present further opportunities for introducing extra functionality.”

Figure R2. FT-IR of MF-1a after treatment of 1M HCl at 100 °C for 1 day. The dotted line indicates the appearance of aldehyde groups from imine bond hydrolysis.

Regarding increasing the conversion rate:

Thanks for the suggestion. We have tried the iterative synthesis as suggested, however, there is no apparent change of the degree of functionalization after two iterations.

2. For the response to comment 2, reviewer 1:

If the author could be able to increase the conversion rate of the Povarov reaction, better quality ^{13}C NMR spectra could be achieved for the post-synthetically modified COFs. It will be better to have a ^{13}C solid state CP-MAS NMR spectra for MF-1d also for the confirmation of the presence of -CN in MF-1d.

Response: The CP-MAS ^{13}C -NMR for MF-1d was acquired and presented in Figure R3. The spectrum shows sufficient S/N ratio and reasonable peak broadness. As can be seen from the spectrum, the quinoline moiety of MF-1d shows broad signals between 100 and 140 ppm that is distinct from the starting material COF-1. Unfortunately, -CN carbon falls within the same region with these aromatic carbons and is impossible to resolve. Despite that, the FT-IR spectrum of MF-1d has unequivocally indicated the presence of CN by the characteristic CN resonance at around 2200 cm^{-1} (Figure R3b). Corresponding changes are made on page 11:

“while solid-state CP-MAS ^{13}C NMR (Figure S18) and...”

Figure R3. (a) Stacked ^{13}C solid-state CP-MAS NMR spectra of COF-1 and MF-1d; (b) FT-IR spectra of COF-1, MF-1d and substrate 1d, 4-ethynyl benzonitrile.

3. For the response to comment 6, reviewer 2:

According to the author, the MF-1 is stable in 12 M HCl at 50 oC and in 14 M NaOH at 60 oC (Table S6). What will happen if the temperature is increased to 100 oC? In case of CAF-1 and CAF-2, they are stable up to 100 oC in HCl and as well as NaOH solution. Rest of the others comments have been addressed by the author suitably.

Response: CAF-1 and CAF-2 (*Nat. Commun.*, **2018**, 8, 1102, Table S6) are stable up to 100 °C for 1 d in 1 M HCl and 1 M NaOH solutions, which are much lower concentration than those in our study (12 M HCl and 14 M NaOH). Following the suggestion of the Reviewer 1, we tested MF-1a under the same conditions and found that MF-1a retained crystalline while COF-1 renders amorphous in HCl (Figure R4a). For the NaOH stability test, both MF-1a and COF-1 show retention of crystallinity after 1d at 100 °C in 1M NaOH, but COF-1 displays more obvious loss in crystallinity. Moreover, the washing solution of COF-1 after treatment turned yellow, an indication of linker leaching from the decomposition of COF-1, while the washing solution of MF-1a remains colorless.

Prompted by the Reviewer, we further tested the chemical stability in boiling concentrated HCl (12 M) and NaOH (14 M) for 1 day. We would like to emphasize that this is, to our knowledge, the first time that any study has tested the stability of a COF in the combination of boiling temperatures AND concentrated acid. Previous studies have used only concentrated acid at room temperature OR dilute acid at boiling temperatures. MF-1a still remained crystalline despite the apparent loss in crystallinity after HCl and NaOH treatment (Figure R4c and R4d) while COF-1 rendered amorphous. Taken together, MF-1a exhibited strikingly high chemical stability in strong acid and base at elevated temperatures (100 °C). Corresponding changes are made on page 10:

“... or to boiling acids (1 M and 12 M HCl) and bases (1 M and 14 M NaOH) for 1 day (Figure S14).”

Figure R4. PXRD patterns of COF-1 and MF-1a at 100 °C for 1d in (a) 1 M HCl; (b) 1 M NaOH; (c) 12 M HCl; (d) 14 M NaOH. Inset are photographs of COF-1 and MF-1a in 1 M HCl and 1 M NaOH at 100 °C for 1d.

4. In addition, one more point should be addressed by the author.

1. For MF-1b the contact angle is 35° (indicating hydrophilic in nature) and for MF-1d the contact angle is 132° (indicating hydrophobic in nature). According to the literature report (*J. Phys. Chem. B* 2013, 117, 7718–7723 & *Phys. Chem. Chem. Phys.*, 2014, 16, 13262–13270) through the dipole moment of $-\text{CN}$ is higher than $-(\text{C}=\text{O})\text{O}^-$ group but their affinity towards water as well as solubility in water are almost same and they are both hydronutral in nature. But the contact angle measurement of the MF-1b and MF-1d contradicts this

Response: We realized that the low contact angle of MF-1b is due to partial hydrolysis of the pendent *methyl ester* groups, which occurred during the workup when MF-1b was treated with a saturated NaHCO_3 solution to remove Lewis catalyst involved in the modification process. Due to the labile chemical activity, $-(\text{C}=\text{O})\text{O}-\text{Me}$ group in MF-1b is prone to undergo partial hydrolysis (incomplete since FT-IR and ^{13}C solid-state NMR spectra clearly show the presence of ester groups) to yield $-\text{COONa}$, imparting hydrophilicity of MF-1b. In a control experiment, we measured the contact angle of MF-1b samples without the treatment of NaHCO_3 and found that the sample is indeed hydrophobic with a contact angle of 122° (See Figure R5a). This phenomenon is even more obvious when we changed the ester from $-(\text{C}=\text{O})\text{O}-\text{Me}$ to $-(\text{C}=\text{O})\text{O}-t\text{Bu}$, which is more prone to hydrolytic cleavage. As shown in Figure R5b, treatment of the *t*-butyl ester bearing MF-1f with NaHCO_3 leads to the extraction of MF-1f from the organic phase into the aqueous phase. The corresponding compressed pellet displays a contact angle of 0°.

(Note: More studies on the new MF-1f and its hydrolyzed products are ongoing and are not included in the current manuscript).

Corresponding changes are made on page 13:

“The methyl ester-bearing MF-1b has a similar contact angle of 122°. Upon treatment with NaHCO₃, the contact angle (MF-1b') decreases significantly to 35°, indicating the feasibility of imparting hydrophilicity via hydrolysis of ester moieties.”

Figure R5. (a) Water contact angles of water droplets on the pressed pellet of COF-1 and MF-1b with/without NaHCO₃ treatment; (b) Water contact angles of water droplets on the pressed pellet of COF-1 and MF-1f. Photograph shows that MF-1f is hydrophilic after the treatment of NaHCO₃.

REVIEWERS' COMMENTS:

Reviewer #1 (Remarks to the Author):

I think this manuscript could be accepted now as the authors have satisfactorily responded to all the comments.